# MRI-Based Assessment of Masticatory Muscle Changes in TMD Patients after Whiplash Injury

**DOI:** 10.3390/jcm10071404

**Published:** 2021-04-01

**Authors:** Yeon-Hee Lee, Kyung Mi Lee, Q-Schick Auh

**Affiliations:** 1Department of Orofacial Pain and Oral Medicine, Kyung Hee University Dental Hospital, #613 Hoegi-dong, Dongdaemun-gu, Seoul 02447, Korea; dental21@khu.ac.kr; 2Department of Radiology, Kyung Hee University College of Medicine, Kyung Hee University Hospital, #26 Kyunghee-daero, Dongdaemun-gu, Seoul 02447, Korea; oralmedicine@khu.ac.kr

**Keywords:** whiplash injury, temporomandibular disorder, magnetic resonance imaging, masticatory muscle, lateral pterygoid muscle, masseter muscle

## Abstract

Objective: to investigate the change in volume and signal in the masticatory muscles and temporomandibular joint (TMJ) of patients with temporomandibular disorder (TMD) after whiplash injury, based on magnetic resonance imaging (MRI), and to correlate them with other clinical parameters. Methods: ninety patients (64 women, 26 men; mean age: 39.36 ± 15.40 years), including 45 patients with symptoms of TMD after whiplash injury (wTMD), and 45 age- and sex-matched controls with TMD due to idiopathic causes (iTMD) were included. TMD was diagnosed using the study diagnostic criteria for TMD Axis I, and MRI findings of the TMJ and masticatory muscles were investigated. To evaluate the severity of TMD pain and muscle tenderness, we used a visual analog scale (VAS), palpation index (PI), and neck PI. Results: TMD indexes, including VAS, PI, and neck PI were significantly higher in the wTMD group. In the wTMD group, muscle tenderness was highest in the masseter muscle (71.1%), and muscle tenderness in the temporalis (60.0%), lateral pterygoid muscle (LPM) (22.2%), and medial pterygoid muscle (15.6%) was significantly more frequent than that in the iTMD group (all *p* < 0.05). The most noticeable structural changes in the masticatory muscles occurred in the LPM with whiplash injury. Volume (57.8% vs. 17.8%) and signal changes (42.2% vs. 15.6%) of LPM were significantly more frequent in the wTMD group than in the iTMD group. The presence of signal changes in the LPM was positively correlated with the increased VAS scores only in the wTMD group (r = 0.346, *p* = 0.020). The prevalence of anterior disc displacement without reduction (ADDWoR) (53.3% vs. 28.9%) and disc deformity (57.8% vs. 40.0%) were significantly higher in the wTMD group (*p* < 0.05). The presence of headache, sleep problems, and psychological distress was significantly higher in the wTMD group than in the iTMD group. Conclusion: abnormal MRI findings and their correlations with clinical characteristics of the wTMD group were different from those of the iTMD group. The underlying pathophysiology may differ depending on the cause of TMD, raising the need for a treatment strategy accordingly.

## 1. Introduction

Whiplash injury refers to a macrotrauma that occurs with a hyperextension of the head and neck vertebrae followed by hyperflexion when a sudden or excessive force is applied [1]. In road traffic accidents, injuries occur mainly due to side-impact or rear-end collisions [2,3] (Figure 1). Such cases present with a variety of clinical manifestations, including neck stiffness, neck pain disability, psychological distress, headache, and temporomandibular disorder (TMD) [4,5,6]. The incidence of TMD in patients with whiplash injury is low to moderate and ranges from 14–37.5% [4]. It is important to note that patients suffering from chronic whiplash injury commonly have clinical pain in a wider range of their bodies [7]. Approximately 40% of patients with whiplash injuries suffer from persistent pain and chronic disability [8].

TMD is an umbrella term for pain and dysfunction in the temporomandibular joint (TMJ), masticatory muscles, and adjacent structures. Population-based studies showed that 60–70% of the general population has at least one sign of TMD, 12.1% have TMD-related pain, and only 5% seek treatment [9,10]. Plausible risk factors for TMD, either alone or more likely in combination include macrotrauma, microtrauma, other body pain conditions, self-reported previous pain experience, and psychological status. The etiology of TMD moved from a mechanical-based phenomenon to a chronic pain biopsychosocial model [11]. Typical and frequent TMD symptoms include TMJ noise and pain, followed by restriction of mandibular movement, headache, neck pain, ear pain, and tinnitus. Whiplash injury is considered an initiating or aggravating factor of TMD [5,12]. Thus, patients with TMD suffering from whiplash injury have wider and stronger pain than those with only microtrauma [6].

The pathophysiology of TMD development and exacerbation associated with whiplash injury can be distinguished from that due to other TMD etiologies. TMD caused by functional habits or microtraumatic accumulations such as malocclusion, i.e., idiopathic causes, are arise from at changes at the peripheral level. However, during whiplash injury, individuals have direct macrotrauma to muscles, ligaments, and tendons of cervical area, and significant damage to surrounding tissues and pain may also occur [13]. In addition, sudden large impact can be transmitted to the central nervous system (CNS) because of the occurrence of diffuse axonal injury throughout the injury [14]. Sometimes referral patterns may develop because of CNS involvement resulting from prolonged masticatory muscle pain, and the individual’s pain pattern becomes more complex. Additionally, experiences of emotional trauma from the injury that pose a potential threat to life can negatively impact personal future health and finances [15]. These physical and psychosocial risk factors create combined synergies and form complex neuropathologies of pain amplification in TMD.

Masticatory muscles are poorly understood in terms of posttraumatic sequelae. All four main masticatory muscles, including the masseter, temporalis, medial pterygoid muscle (MPM), and lateral pterygoid muscle (LPM), are attached to the mandible and govern the function to move the mandible (Figure 2). The masseter and temporalis muscles on the outside of the jaw bones have been studied, and it has been reported that these muscles are commonly affected in patients with TMD [16]. The masseter is a jaw-closing muscle that elevates the mandible and is one of the strongest in the human body. The temporalis muscle enables elevation and retraction of the mandible. An important role of the MPM is the fine control and stabilization of the vertical mandibular position, concentrically activating jaw closing [17]. Among all four muscles of mastication, the LPM is the only muscle that depresses the mandible. Traumatic injury results in muscle pain, muscle wasting, skeletal demineralization, and disability [18]. However, there is limited information on the changes in the masticatory muscles of patients with TMD after whiplash injury.

Magnetic resonance imaging (MRI) is the gold standard for evaluating the abnormalities in the TMJ and masticatory muscles, and for determining the disc–condyle relationship [6,12]. Atrophy, fibrotic change, and fatty replacement of the masticatory muscle regularly occur with either disuse or denervation [19], which can occur as a consequence of immobilization or disuse after whiplash injury. On T1- and T2-weighted MRI images, masticatory muscles with myotonic dystrophy present as high signal intensity areas because of fatty replacement [6,20]. In addition, the huge tensile force throughout the whiplash injury exerted on a muscle may lead to excessive stretching of the muscle fibers and consequently a tear, and impair the muscle-tendon junction [21]. According to our previous MRI-based studies, patients with TMD due to whiplash injury, suffered more distortion of the disc–condyle relationship and LPM changes [6,12]. TMJ disc displacement leads to degenerative changes in the disc deformity and condylar degeneration [22], and the symptoms of TMD might become chronic. Until now, most MRI studies in patients with TMD evaluated intra-articular changes [23].

Thus, the present study aimed to investigate the relationship between changes in the masticatory muscles and TMJ on MRI and clinical pain intensity in patients with TMD due to whiplash injury, and to compare their clinical and MRI characteristics with those of patients with TMD due to idiopathic causes. We also investigated the factors that affect subjective pain intensity and whether these results differ between the TMD due to idiopathic causes (iTMD) and TMD after whiplash injury (wTMD) groups. Our hypotheses were as follows: (1) compared to the patients with iTMD, patients with wTMD will have higher pain intensity and more changes in the masticatory muscles, and have more tenderness upon palpation; (2) structural changes in the TMJ and masticatory muscles will be associated with a higher severity of TMD symptoms. Our study is the first study to examine the changes in the four major masticatory muscles in patients with whiplash injury using MRI, and to determine which muscular factors increase discomfort and pain in patients with wTMD.

## 2. Materials and Methods

### 2.1. Participants and Demographic Data

This retrospective case-control study involved 45 randomly selected patients with wTMD and 45 randomly selected patients with iTMD, who were selected using a simple random sampling procedure employing a random number table. The “wTMD” group (*n* = 45) included patients who had first-onset TMD after sustaining a whiplash injury; the “iTMD” group (*n* = 45) comprised patients who had presented with idiopathic or non-traumatic TMD symptoms without any history of head/neck trauma.

The patients were diagnosed with TMD using the research diagnostic criteria (RDC) for TMD axis I [24,25] and the experience of whiplash injury was judged according to the Quebec Task Force on Whiplash-Associated Disorders [2]. We identified the patients who had TMD and retrospectively reviewed all MRIs of their TMJs and TMJ reports from March 2017 through 2020. MRIs for patients with TMD were taken during the TMD diagnosis phase, between the first and second hospital visits. The patients with any level of data loss were excluded from the study. The inclusion criteria were as follows: no history of neck pain prior to the whiplash injury, no treatment on the current TMD symptoms other than medication, and no history of direct trauma to the jaw before or during the accident—the patients had no history of a TMD prior to the present TMD episode. The exclusion criteria were serious injuries, such as facial fracture and unstable multiple traumas, previous injury, neurological disorder unrelated to the trauma, and musculoskeletal disorder predating injury, rheumatism, psychological problems, and pregnancy. To assess the impact of whiplash injury in terms of distribution of TMD signs and symptoms, and MRI findings, all variables were compared by group.

### 2.2. MRI Acquisition and Analysis

All 90 patients underwent MRI examination on bilateral TMJs and masticatory muscles. High-resolution MRI images were obtained on a 3T MRI system (Signa™ Genesis, GE Healthcare, Chicago, IL, USA) with a 6-cm × 8-cm diameter surface coil. MRI examination was dependent on the MR sequences and protocol of Kyung Hee University Medical Center, and the details has also been revealed in our previous studies [6,12]: All scans involved sagittal oblique sections (section thickness, 3 mm or less; field of view, 15-cm; matrix dimensions, 256 × 224 matrix) and spin-echo sagittal MR images were obtained on the axial localizer images. T1-weighted images (T1WIs) and T2-weighted images (T2WIs) were obtained using a 650/14 repetition time (TR)/echo time (TE) sequence and 2650/82 TR/TE sequence, respectively. The proton density images were obtained using a 2650/82 TR/TE sequence.

TMJs and adjacent masticatory muscles were evaluated in both the sagittal and coronal planes (Figure 3) to determine the abnormalities in the TMJ and the alteration of each masticatory muscle. Two head and neck radiologists with at least 8 years of experience in their field, and who were blinded to the patients’ clinical information, visually analyzed the MRI findings. After determining the existence of MRI abnormalities, the TMJ and masticatory muscle measurements were evaluated using the INFINITT PACS (INFINITT Corp., Seoul, Korea).

### 2.3. Validation of MRI Findings

We investigated the presence of MRI abnormalities of the TMJ and masticatory muscles, including disc displacement, effusion, disc deformity, condylar degeneration, and volume changes (VCs) and signal changes (SCs) in the four masticatory muscles, including the masseter muscle, temporalis muscle, LPM, and MPM. The detailed procedure of validation has been described in our previous papers [6,12]. The disc position in the oblique sagittal plane was determined in the closed and open-mouth positions. In addition, disc position was classified as anterior disc displacement with reduction (ADDWR) or without reduction (ADDWoR) (Figure 4). In the T1WI, T2WI, and proton density (PD) weighted images, each muscle was considered to have an SC when the fatty replacement tissue accompanying the high intensity signal was observed widely across the muscle. In the same images, it was considered to have a VC when the masticatory muscle had decreased volume and fibrosis. In the same images, it was considered to have a VC when the masticatory muscle had decreased volume and fibrosis, which had the muscle contracture with low-density signal. To determine whether the difference between the measurements was statistically significant, the values of the masticatory muscles acquired in each MRI sequence on the right and left sides of the same patient were compared.

### 2.4. Validation of Clinical Signs and Symptoms of TMD

#### 2.4.1. Contributing Factors and Comorbidities

We investigated self-reported parafunctional activities using the Oral Behavior Checklist, which includes jaw-related behaviors, such as teeth clenching and bruxism [26]. Self-reported headache was evaluated using the dichotomous question “Do you have any headaches associated with TMD?” Self-assessment of tinnitus, psychological distress, and sleep problems was also reported. Each parameter was recorded as a binary answer (yes/no) in all patients.

#### 2.4.2. Characteristics of Pain

The duration of pain derived from the TMJ and adjacent structures was reported in days. Temporomandibular pain was scored by the patients subjectively, ranging from 0 (no pain at all) to 10 (the worst pain imaginable) using the visual analog scale (VAS).

#### 2.4.3. Palpation Index (PI) and Neck PI

PI is a reliable scoring system that analyzes the severity of myofascial pain, allowing TMD symptoms to be objectively evaluated. In each patient, we palpated 20 intra- and extraoral muscle sites and three sites in the neck. The index-finger palpation pressure was 1 kg/cm^2^ and was applied for 5 s, as recommended in previous studies [27,28]. To calibrate palpation pressures, we regularly pre-tested our index-finger pressure using a hand-held pressure algometer. For each site, a binary answer (yes/no) was provided. To calculate the PI score, we added all the positive answers and divided the sum by the number of events [29]. To further investigate the intensity of neck pain, we calculated the neck PI, which was defined as the number of positive responses to the palpation of neck muscles, including the sternocleidomastoid, splenius capitis, and trapezius muscles, divided by the number of events. Using these two indexes, we quantified clinical myofascial pain.

### 2.5. Statistical Methods

For all discrete variables, the absolute and relative percentage distributions were calculated. Continuous variables are presented as means and standard deviations. Various statistical methods were used for data analysis. Using the Mann–Whitney U test, we examined the difference in the means of the continuous variables between the wTMD and iTMD groups. Chi-square and Fisher’s exact tests were used to calculate the proportional equivalence of discrete variables, such as the presence of MRI abnormalities (%) and patient’s “yes” (%) response describing TMD contributing factors and comorbidities. To measure inter-rater reliability in the presence or absence of abnormalities in MRI findings, we used Cohen’s kappa coefficient. Through Spearman’s correlation analysis, we examined the MRI findings that were correlated with a significant increase in each TMD index score. A *p* < 0.05 level was set to be statistically significant. Data were analyzed using the Statistical Package for the Social Sciences for Windows version 25.0 (IBM Corp., Armonk, NY, USA).

## 3. Results

### 3.1. Demographic Patient Characteristics

Data from 45 patients with wTMD (mean age: 37.94 ± 12.27) and 45 patients with iTMD (mean age: 40.77 ± 18.02 years) were analyzed and compared. Among the patients with TMD who visited the hospital during the study period, female predominance (female:male = 2.45:1) was observed, and this sex ratio was reflected in both the wTMD (female: 68.9%) and iTMD groups (female: 73.3%). In addition, the mean symptom duration of wTMD (66.27 ± 58.06 days) was not different from that of iTMD (57.78 ± 48.33 days) (*p* = 0.453).

### 3.2. Analysis of Pain Intensity

The mean VAS score in the wTMD group was significantly higher than that in the iTMD group (6.73 ± 1.84 vs. 4.32 ± 2.80, *p* < 0.001). Interestingly, TMD indexes, including PI and neck PI, were significantly higher in the wTMD group than in the iTMD group (Table 1). That is, the mean PI (0.27 ± 0.21 vs. 0.16 ± 0.18, *p* < 0.01) and neck PI (0.45 ± 0.31 vs. 0.27 ± 0.23, *p* < 0.01) scores in the wTMD group were significantly higher than that in the iTMD group.

### 3.3. Clinical Characteristics in wTMD and iTMD Groups

Table 2 shows the putative contributing factors and clinical symptoms of TMD in each group. The distributions of clenching habits, headache, sleep problems, and psychological distress in the wTMD group were different from those in the iTMD group. Specifically, the patients with wTMD were more likely to have headache (71.1% vs. 40.0%) and sleep problems (57.8% vs. 35.6%), and psychological distress (33.3% vs. 13.3%), whereas clenching habits (8.9% vs. 26.7%) were less common in the wTMD group (all *p* < 0.05). The distributions of bruxism and tinnitus in the wTMD group were not significantly different from those in the iTMD group.

### 3.4. Distribution of MRI Findings

The distributions of MRI variables on the TMJ (Table 3) and masticatory muscles (Table 4) were significantly different between the wTMD and iTMD groups. Cohen’s kappa coefficients for all factors showed excellent agreement (0.88 to 0.93). In the case of disagreement between the two experts, MRI readings were confirmed after discussion. Interestingly, patients with wTMD were more likely to have ADDWoR (53.3% vs. 28.9%) and disc deformity (64.4% vs. 33.3%), which were significantly higher than those in the iTMD group (all *p* < 0.05). The distributions of effusion, ADDWR, and condylar degeneration in the wTMD group were not significantly different from those in the iTMD group. The most prominent abnormal MRI finding in the wTMD group was disc deformity (64.4%), followed by condylar degeneration (57.8%), ADDWoR (53.3%), ADDWR (24.4%), and effusion (22.2%) (Figure 5).

Regarding the masticatory muscles, the structural change in LPM was the most frequent change among the four major masticatory muscles in both groups (Figure 6). Specifically, both VC (57.8% vs. 17.8%) and SC (42.2% vs. 15.6%) of the LPM were significantly more prevalent in the wTMD group than in the iTMD group (all *p* < 0.05). In contrast, VC (2.2% vs. 2.2%) and SC (2.2% vs. 2.2%) of the temporalis muscle were very rare in both groups. In addition, changes in MPM were not observed in any case in the wTMD group, and both VC and SC were observed in only two patients (4.4%) in the iTMD group. The structural changes in the masseter muscle on MRI were not observed in any case (0.0%) in either group (Figure 7).

### 3.5. Distribution of Muscle Tenderness

The distributions of muscle tenderness on palpation test were significantly different between the groups in the temporalis muscle, LPM, and MPM except for the masseter muscle (Table 5). Muscle tenderness was significantly higher in the temporalis muscle (60.0% vs. 35.6%), LPM (22.2% vs. 2.2%), and MPM (15.6% vs. 0.0%) in the wTMD group than in the iTMD group (all *p* < 0.05). On the other hand, tenderness was most frequently observed in the masseter muscle in both wTMD (71.1%) and iTMD (64.4%) groups. In contrast to the remarkable frequent observations of actual structural changes in the LPM observed on MRI, muscle tenderness of the LPM was scarcely observed in the iTMD group (2.2%) and only in 22.2% of patients in the wTMD group.

### 3.6. Associated MRI Findings with Increasing TMD Index

VAS, which represents the degree of subjective pain, was positively correlated with masticatory muscle changes in wTMD, and with internal derangement of the TMJ in iTMD. Specifically, the presence of signal changes in the LPM was positively correlated with the increased VAS scores only in the wTMD group (r = 0.346, *p* = 0.020). In contrast, the presence of ADDWoR (r = 0.325, *p* = 0.029) and condylar degeneration (r = 0.297, *p* = 0.047) was positively correlated with the increase in VAS scores in the iTMD group. In the iTMD group, the VAS score was moderately positively correlated with PI (r = 0.300, *p* = 0.045) and neck PI (r = 0.301, *p* = 0.044), and these correlations were not observed in the wTMD group. Regarding PI and neck PI scores, there were no significant MRI variables that were correlated. However, neck PI and PI were strongly positively correlated with each other in both the wTMD (r = 0.866, *p* < 0.001) and iTMD (r = 0.764, *p* < 0.001) groups (Table 6).

### 3.7. Diagnostic Classification of Pain-Related TMD

Depending on the cause of TMD pain, the most common pain-related diagnostic criteria for TMD fall into two categories: muscular and joint pain. Pain of muscular origin includes subcategories of myofascial pain and myofascial pain with limited mouth opening. On the other hand, pain of joint origin means arthralgia. In this study, the TMD diagnosis was established according to RDC/TMD, and TMD was diagnosed into three subgroups [25]: pain of muscle origin, pain of joint origin, and mixed type. Because 93% of patients with wTMD and 89% of iTMDs were diagnosed with TMD with mixed muscle and joint pain, the data were not analyzed based on subgroups.

## 4. Discussion

Whiplash, a unique type of macrotrauma, mainly occurring in vehicle collisions, and is caused by sudden dynamics of hyperextension-hyperflexion on the cervical spine [30]. Among the patients with whiplash injury, approximately 23% suffered chronic pain and disabilities [31]. Thus, it lowers the patient’s quality of life and increases health costs. Neck pain and stiffness, and headache, are the most prominent symptoms in whiplash injury [6]. Recently, a possible association between whiplash injury and TMD has been studied. Carroll et al. reported that TMD was more prevalent in individuals with whiplash injury (15.8%) than in those without (4.7%) [32]. Conversely, the prevalence of whiplash injury in populations with TMD (median 35%) is higher than that in control groups without TMD (1.7–13%) [5]. Although there is limited information on the neuropathophysiology of TMD development and aggravation after whiplash injury, research to explore this is ongoing. As hypothesized, many of the clinical symptoms and MRI findings of masticatory muscles of whiplash-related TMD can be distinguished from those of TMD resulting from an unknown cause.

As in our first hypothesis, patients with wTMD had higher pain intensity across wider jaw and neck areas compared to those with iTMD. This was consistent with previous results that patients with whiplash injury report higher pain scores and larger areas of local and referred pain than healthy controls [33]. The increased clinical pain observed in the wTMD group can be related to nociplastic pain, which implies changes in function of nociceptive pathways [34]. This pain pattern can be caused by insufficient or impaired diffuse noxious inhibitory controls (DNICs), a measure of central nervous system pain inhibition [35]. Moreover, sleep disturbance is related to decreased DNICs in patients with TMD [36]. Reduced reactivity of the hypothalamic–pituitary adrenal (HPA) axis has been associated with chronic widespread body pain [37]. Furthermore, thalamic activity also contributes significantly to the pain processing. Decreased thalamic activity can cause exaggerated pain following innocuous peripheral stimulation [38]. Collectively, changes to the process of central endogenous pain inhibition through interference with DNICs, HPA axis, or thalamic activity can occur in patients with whiplash-related TMD, and pain is more likely to be amplified and prolonged in such patients.

Widespread pain, which is not limited to the injured area and increased pain intensity, is derived from the dynamics of the nociceptive pathway at the CNS level. Hypersensitivity after whiplash injury occurs both locally, i.e., throughout the neck area, and at more distant sites beyond the boundary of the damaged peripheral nerve [39]. Thus, deep tissue is damaged in the cervical joint by direct application of shear force, compression, and excessive stretching through whiplash injury, but it can also affect the TMJ area. Pain wind-up refers to the phenomenon of increased excitability of CNS induced by frequency-dependent electrical stimulation of afferent C-fibers [40]. Pain wind-up can also be a mechanism of hypersensitivity and in whiplash-associated TMD. Patients with painful TMD have reduced pain thresholds and sensory impairment after innoxious stimuli [39]. Interestingly, central sensitization is not only specific to whiplash injury, even though this phenomenon is seen in a variety of chronic pain syndromes [39,41]. Similar pathophysiology may underlie a variety of chronic pain, including TMD, and further investigation is required on how it serves in detail.

Regarding MRI abnormalities, the wTMD group had more prevalent changes in the structure of TMJ and masticatory muscles than the iTMD group. Specifically, more than 50% of patients with wTMD had ADDWoR and condylar degeneration, and were significantly more prevalent than those with iTMD. In general, the TMJ is the most affected structure from TMD [42]. To the best of our knowledge, only a few MRI studies have validated the signs and symptoms of whiplash injury-induced TMD. During sudden macrotrauma, abrupt changes in the position of the mandible and TMJ disc followed by TMJ-ligament elongation and TMJ disc displacement may occur [43]. In a previous MRI study, displacement (56%) and abnormal joint fluid or edema (65%) of the TMJ were observed in the patients who had TMD after sustaining whiplash injury [44]. In addition, proper disc positioning is a prerequisite for normal movement of the mandibular condyle. Abnormal positioning of the articular disc can evoke TMJ clicking and pain, can lead to triggering a closed locking jaw position, or may as well as limit jaw function [45]. Disc displacement progresses from reducible to non-reducible, and ADDWoR and condylar degeneration can be associated with each other [46].

Considering the masticatory muscle abnormalities observed on MRI, structural changes were also observed more frequently in wTMD. In particular, patients with wTMD had remarkable structural changes in the LPM. Substantial structural change in the LPM may have been observed considerably more frequently in wTMD than in iTMD because LPM is more susceptible to whiplash injury than other masticatory muscles. Additionally, LPM is a direct major factor in the occurrence of whiplash injury-related TMD. The LPM controls the rotation and translation of the disc and condyle, protrudes the mandible, and stabilizes the articular disc [47]. LPM plays a somewhat secondary role in mastication, but it is directly related to changes in the mandibular condyle and disc. Pathologic changes in the LPM can be associated with TMJ disc displacement [48,49]. Recent MRI results have shown structural muscle changes in the form of reduced muscle volume, fatty infiltration, and muscle atrophy in patients with whiplash-related disorders [50,51]. In our previous LPM-related MRI study, significant positive correlations were reported between structural changes in the LPM, ADDWoR, disc deformity, and condylar degeneration in patients with TMD after whiplash injury [6].

Among and/or within the human masticatory muscles, many anatomical differences exist, indicating that different muscles are specialized for their resistance against whiplash injury. In our results, more than 60% of patients with wTMD had muscle tenderness in the masseter and temporalis muscles. Architecturally, the masseter and temporalis muscles can deliver higher forces than the pterygoid muscles. Thus, masseter and temporalis muscles play a more crucial role in mastication, which can lead to fatigue build-up and become vulnerable to tenderness [52]. In addition, the average thickness values of the masseter muscle (13.65 ± 2.19 mm), temporalis muscle (6.66 ± 1.14 mm), MPM (14.73 ± 1.32 mm), and LPM (15.59 ± 1.40 mm) were different [53,54,55]. Specialization in fiber type composition and fiber cross-sectional area can be reflected in these intramuscular differences. Compared to the masseter and pterygoid muscles, the temporalis has significantly larger fibers and a notably different fiber type composition [56]. Furthermore, the myosin heavy chain (MyHC) content of muscle fibers mainly determines their force–velocity properties [57]. A regionally higher proportion of MyHC type I, expressed in slow muscle fibers, was found in the anterior temporalis, deep masseter, and anterior MPM [58]. However, a previous study showed that MyHC type and isoform composition do not sufficiently explain the difference between the form and function of the muscles [59]. Further research is needed to reach a clearer conclusion about the impact of whiplash-related TMD on each masticatory muscle.

As in our second hypothesis, structural changes in the TMJ and masticatory muscles were correlated with symptom severity of TMD. Interestingly, the MRI abnormalities correlated with the VAS score showed differences between groups. The VAS score, indicating the patient’s subjective pain level, was positively correlated with ADDWoR and disc deformity only in iTMD, and this correlation was not observed in wTMD. Actually, the articular disc displacement with reduction is commonly asymptomatic and does not require specific treatment, since the TMJ structure adapts well and painlessly to the changed disc position [60]. However, displaced disc position is considered a putative risk factor for TMJ degenerative joint disease and progression to ADDWoR [61]. ADDWoR can lead to TMJ pain and painful locking [62]. The limited mouth opening, inflammation, and pain in their TMJ and surrounding structures are common causes that individuals with iTMD come to the hospital for the treatment. We also found that the VAS score was correlated with the structural changes of LPM only in wTMD. Rapid LPM stretching during macrotrauma induces reflex contracture with disc displacement, resulting in pain [6,63]. On the contrary, no muscle changes were observed in asymptomatic individuals or individuals with idiopathic or non-traumatic neck pain [50]. Having pain even with small structural changes in iTMD may suggest that both nociceptive and nociplastic pain mechanisms may be potentially involved as symptoms become chronic [64]. Therefore, the major biostructural factors contributing to the pain of wTMD may be different from those of the iTMD.

Finally, we observed that headache, sleep problems, and psychological distress were significantly more prevalent in the wTMD group than in the iTMD group. Headaches, sleep problems, and psychological distress may be related to CNS pathology rather than peripheral pathologies [65,66,67]. We must focus on the fact that comorbidities associated with CNS involvement were more prevalent in the wTMD group. Headache and sleep problems possibly co-occur from the shared underlying mechanisms result of reducing nociceptive activity in trigeminal nucleus caudalis or dysfunctional hypothalamic activity [68,69]. Sleep problems deleteriously affect central pain modulatory systems [36]. Headache has been suggested as an aggravating and potential risk factor for TMD symptoms [70]. Psychological distress is often accompanied by CNS-level symptoms, and is associated with more pain and disability in whiplash injury [41]. In addition, psychological factors, along with physical factors, can play a role in the progress or recovery from whiplash injury [71]. This suggests that neuropathophysiology of wTMD may differ from that of iTMD, and that TMD should be understood in a biopsychosocial model considering macrotrauma. The interconnectivity between biological factors and psychological factors can involve the development, processing, and chronicity of whiplash-related TMD symptoms. Furthermore, we need to have specific coping and treatment strategies for patients with wTMD.

This study has several limitations. In our study, we only checked for the presence of self-reported psychological distress, not through elaborate questionnaires or diagnostic tests. To understand whiplash injury-related TMD in the biopsychosocial model, further systematic investigation of the psychological aspects of patients will be required. Axis II of the DC/TMD or RDC/TMD diagnostic tool might be helpful in examining psychosocial factors [28,72]. In addition, this study has the advantage that it had a retrospective case-control study, but a large-scale population-based study is needed to avoid bias due to data composition and to reach a more general conclusion. The masticatory muscles function cooperatively and elaborately during jaw movement [73]. In addition, the functional association between the masticatory muscles or between masticatory muscles and cervical muscles is mediated by complex central neural processes within the brainstem trigeminocervical complex and not a simple biomechanical coupling [74,75]. Therefore, it is necessary to further study the differences before and after whiplash injury of each muscle over a long period of observation, and the relationships among the masticatory muscles.

## 5. Conclusions

We investigated and analyzed the clinical and MRI findings of patients with TMD caused by whiplash injury, and compared the results of patients with idiopathic causes to TMD. This is the first study to comprehensively investigate the four major masticatory muscles after whiplash injury using MRI, and their correlation with clinical TMD indexes.

We suggest that patients with TMD who have whiplash macrotrauma can have differentiated pain symptomatology from those with idiopathic causes, having more pain in a wider jaw and neck areas, and experiencing more volume and signal changes in the masticatory muscle. Although consistent results for drawing a general conclusion are limited, we hope our results can help in understanding the clinical symptoms and pathophysiology of whiplash-related TMD, and provide information for the appropriate wTMD-specific treatment strategy.

## Figures and Tables

**Figure 1 jcm-10-01404-f001:**
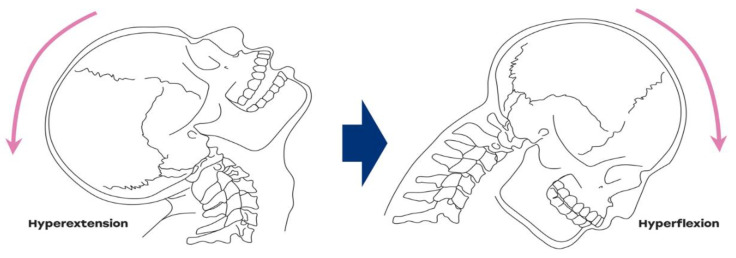
Dynamic mechanism of whiplash injury with hyperextension–hyperflexion. Whiplash injury refers to a macrotrauma that occurs with a sudden hyperextension followed by hyperflexion, which can initiate or aggravate the temporomandibular disorder (TMD) symptoms.

**Figure 2 jcm-10-01404-f002:**
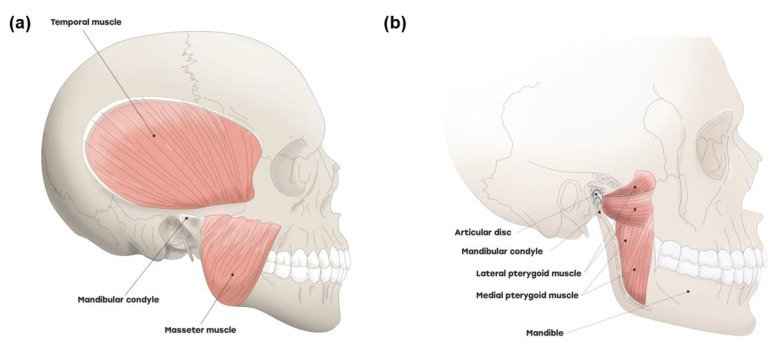
Four main masticatory muscles and temporomandibular joint (TMJ). TMJ and four major masticatory muscles including (**a**) temporalis muscle, masseter muscle, (**b**) lateral pterygoid muscle, and medial pterygoid muscle.

**Figure 3 jcm-10-01404-f003:**
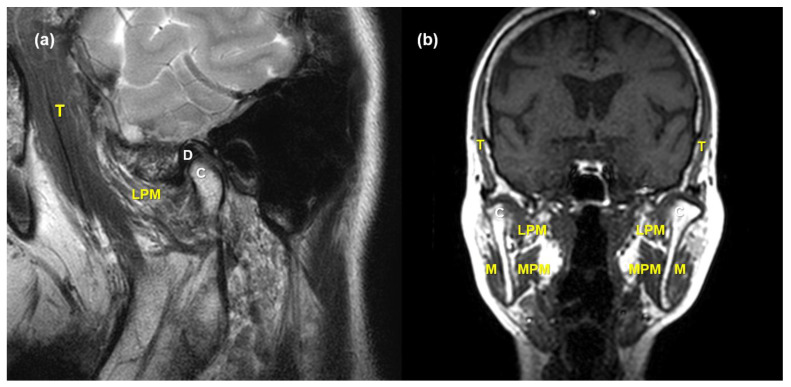
Representative magnetic resonance (MR) images of masticatory muscle and TMJ. T2-weighted (**a**) sagittal and (**b**) coronal MR images represents the location of four main masticatory muscles, articular disc, and mandibular condyle. M, masseter muscle; T, temporalis muscle; LPM, lateral pterygoid muscle; MPM, medial pterygoid muscle; C, condyle; D, articular disc.

**Figure 4 jcm-10-01404-f004:**
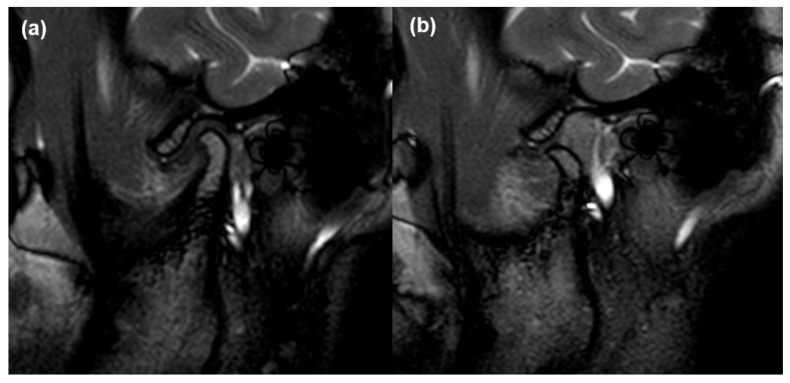
Disc displacement in TMJ on magnetic resonance images. (**a**) Anteriorly displaced articular disc in the oblique sagittal T1-weighted image obtained in the closed-mouth position, and (**b**) anterior disc displacement without reduction where the relationship between disc–condyle is not recaptured in the open-mouth view.

**Figure 5 jcm-10-01404-f005:**
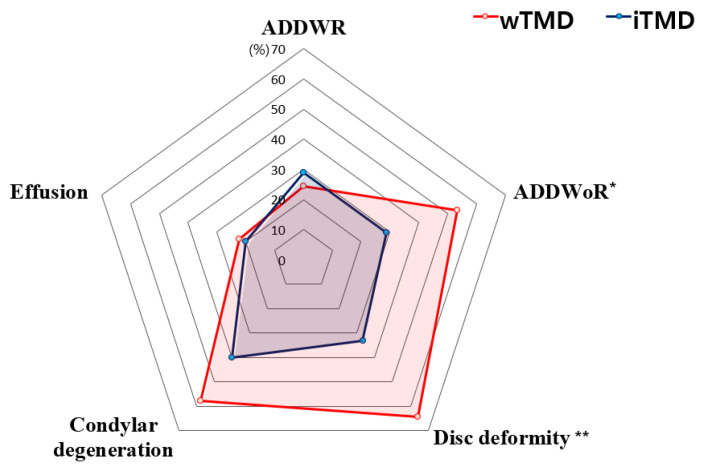
Changes in TMJ from magnetic resonance image. A *p*-value < 0.05 was considered significant. *: *p*-value < 0.05, **: *p*-value < 0.01. ADDWR, anterior disc displacement with reduction; ADDWoR, anterior disc displacement without reduction.

**Figure 6 jcm-10-01404-f006:**
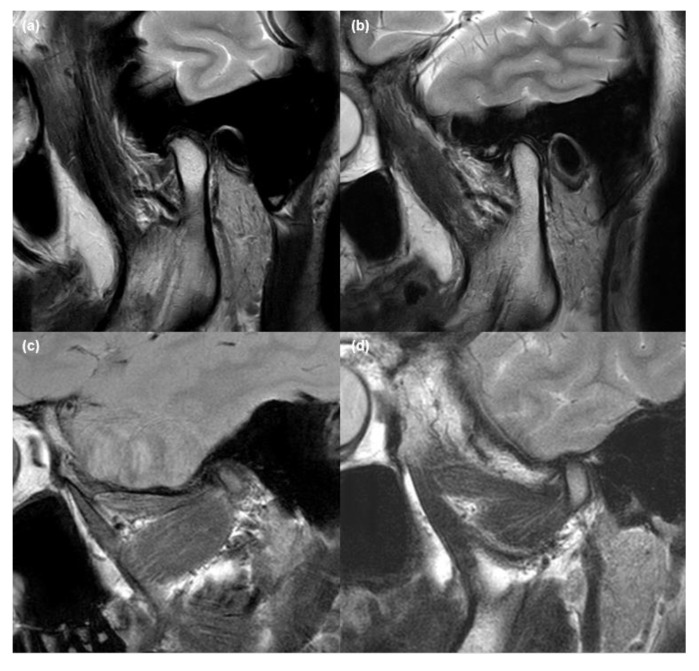
Representative volume changes (VCs) and signal changes (SCs) in the lateral pterygoid muscle (LPM) compared to healthy LPM. (**a**) Proton density-weighted magnetic resonance image obtained in the closed-mouth position showing fatty replacement and atrophic change in the LPM. (**b**) Both the VCs and SCs in the LPM were observed with disc displacement in the closed-mouth position. Representative proton density-weighted magnetic resonance images of healthy LPM without undergoing structural changes (**c**,**d**).

**Figure 7 jcm-10-01404-f007:**
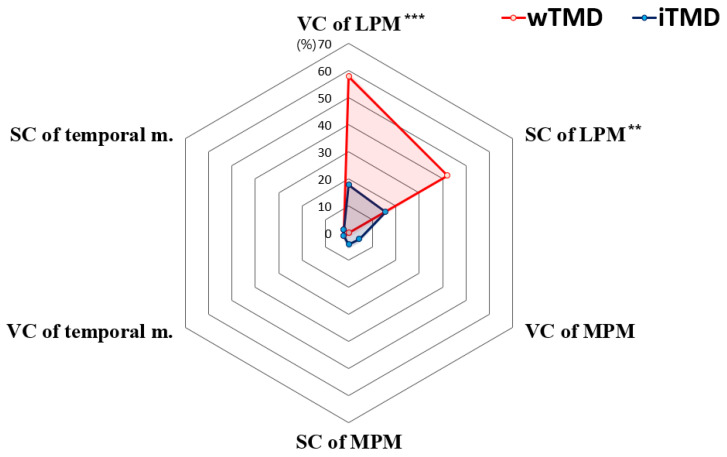
Changes in masticatory muscle abnormalities from magnetic resonance image. A *p*-value < 0.05 was considered significant. *: *p*-value < 0.05, **: *p*-value < 0.01. VC, volume change; SC, signal change; LPM, lateral pterygoid muscle; MPM, medial pterygoid muscle.

**Table 1 jcm-10-01404-t001:** Comparison of demographics, and VAS and TMD index scores.

	wTMD Group Mean ± SD or *n* (%) (*n* = 45)	iTMD Group Mean ± SD or *n* (%) (*n* = 45)	*p*-Value
*Demographics*			
Age, years ^a^	37.94 ± 12.27	40.77 ± 18.02	0.387
Female ^b^	31 (68.9%)	33 (73.3%)	0.816
*Pain characteristics*			
Symptom duration, days ^a^	66.27 ± 58.06	57.78 ± 48.33	0.453
VAS ^a^	6.73 ±1.84	4.32 ± 2.80	**<0.001 *****
*TMD indexes*			
PI ^a^	0.27 ± 0.21	0.16 ± 0.18	**0.0086 ****
Neck PI ^a^	0.45 ± 0.31	0.27 ± 0.23	**0.003 ****

TMD, temporomandibular disorder; wTMD, the patients had experienced whiplash injury and had no TMD symptoms before the injury; iTMD, the patients who presented with idiopathic/non-traumatic TMD symptoms without any history of head/neck trauma; VAS, visual analog scale; PI, palpation index; SD, standard deviation. TMD, temporomandibular disorder; VAS, visual analog scale; PI, palpation index; SD, standard deviation; ^a^: Results were obtained via Mann–Whitney U test; ^b^: chi-square test (two-sided). A *p*-value < 0.05 was considered significant. **: *p*-value < 0.01, ***: *p*-value < 0.001. Significant results are in bold text.

**Table 2 jcm-10-01404-t002:** Distribution of contributing factors and comorbidities of TMD.

	wTMD Group		iTMD Group		
	*n* = 45	Column (%)	*n* = 45	Column (%)	*p*-Value
Bruxism ^a^					
No	40	(88.9)	40	(88.9)	1.000
Yes	5	(11.1)	5	(11.1)	
Clenching ^b^					
No	41	(91.1)	33	(73.3)	**0.026 ***
Yes	4	(8.9)	12	(26.7)	
Tinnitus ^a^					
No	30	(66.7)	35	(77.8)	0.347
Yes	15	(33.3)	10	(22.2)	
Headache ^a^					
No	13	(28.9)	27	(60.0)	**0.006 ****
Yes	32	(71.1)	18	(40.0)	
Sleep problem ^a^					
No	29	(64.4)	19	(42.2)	**0.028 ***
Yes	26	(57.8)	16	(35.6)	
Psychological distress ^a^					
No	39	(86.7)	30	(66.7)	**0.045 ***
Yes	15	(33.3)	6	(13.3)	

TMD, temporomandibular disorder, wTMD, the patients had experienced whiplash injury and had no TMD symptoms before the injury, iTMD; the patients who presented with idiopathic or non-traumatic TMD symptoms without any history of head/neck trauma. The results were obtained via a chi-square test (two-sided), and b: Fisher’s exact test. A *p*-value < 0.05 was considered significant. *: *p*-value < 0.05, **: *p*-value < 0.01. Significant results are in bold text.

**Table 3 jcm-10-01404-t003:** Comparison of joint abnormalities from MRI.

		wTMD Group		iTMD Group		
		*n* = 45	Column (%)	*n* = 45	Column (%)	*p*-Value
ADDWR	No	34	(75.6)	32	(71.1)	0.812
	Yes	11	(24.4)	13	(28.9)	
ADDWoR	No	21	(46.7)	32	(71.1)	**0.032 ***
	Yes	24	(53.3)	13	(28.9)	
Disc deformity	No	16	(35.6)	30	(66.7)	**0.006 ****
	Yes	29	(64.4)	15	(33.3)	
Condylar degeneration	No	19	(42.2)	27	(60.0)	0.140
	Yes	26	(57.8)	18	(40.0)	
Effusion	No	35	(77.8)	36	(80.0)	1.000
	Yes	10	(22.2)	9	(20.0)	

TMD, temporomandibular disorder; wTMD, the patients had experienced whiplash injury and had no TMD symptoms before the injury; iTMD, the patients who presented with idiopathic or non-traumatic TMD symptoms without any history of head/neck trauma; ADDWR, anterior disc displacement with reduction; ADDWoR, anterior disc displacement without reduction. The results were obtained via the chi-square test (two-sided). A *p*-value < 0.05 was considered significant. *: *p*-value < 0.05, **: *p*-value < 0.01. Significant results are in bold text.

**Table 4 jcm-10-01404-t004:** Comparison of masticatory muscle abnormalities from MRI.

			wTMD Group		iTMD Group		
			*n* = 45	Column (%)	*n* = 45	Column (%)	*p*-Value
Masseter muscle ^a^	VC	No	45	(100.0)	45	(100.0)	
		Yes	0	(0.0)	0	(0.0)	
	SC	No	45	(100.0)	45	(100.0)	
		Yes	0	(0.0)	0	(0.0)	
Temporalis muscle ^a^	VC	No	44	(97.8)	44	(97.8)	1.000
		Yes	1	(2.2)	1	(2.2)	
	SC	No	44	(97.8)	44	(97.8)	1.000
		Yes	1	(2.2)	1	(2.2)	
LPM ^b^	VC	No	19	(42.2)	37	(82.2)	**<0.0001 *****
		Yes	26	(57.8)	8	(17.8)	
	SC	No	26	(57.8)	38	(84.4)	**0.010 ****
		Yes	19	(42.2)	7	(15.6)	
MPM ^a^	VC	No	45	(100.0)	43	(95.6)	0.494
		Yes	0	(0.0)	2	(4.4)	
	SC	No	45	(100.0)	43	(95.6)	0.494
		Yes	0	(0.0)	2	(4.4)	

TMD, temporomandibular disorder; wTMD, the patients had experienced whiplash injury and had no TMD symptoms before the injury; iTMD, the patients who presented with idiopathic/non-traumatic TMD symptoms without any history of head/neck trauma; LPM, lateral pterygoid muscle; MPM, medial pterygoid muscle; VC, volume change; SC, signal change. The results were obtained via ^a^: Fisher’s exact test, and ^b^ chi-square test (two-sided). A *p*-value < 0.05 was considered significant. **: *p*-value < 0.01, ***: *p*-value < 0.001. Significant results are in bold text.

**Table 5 jcm-10-01404-t005:** Comparison of muscle tenderness upon palpation test.

	wTMD Group		iTMD Group		
	*n* = 45	Column (%)	*n* = 45	Column (%)	*p*-Value
**Masseter muscle tenderness ^a^**					
No	13	(28.9)	16	(35.6)	0.652
Yes	32	(71.1)	29	(64.4)	
**Temporalis muscle tenderness ^a^**					
No	18	(40.0)	29	(64.4)	**0.034 ***
Yes	27	(60.0)	16	(35.6)	
**LPM tenderness ^b^**					
No	35	(77.8)	44	(97.8)	**0.007 ****
Yes	10	(22.2)	1	(2.2)	
**MPM tenderness ^b^**					
No	38	(84.4)	45	(100.0)	**0.012 ***
Yes	7	(15.6)	0	(0.0)	

TMD, temporomandibular disorder; wTMD, the patients had experienced whiplash injury and had no TMD symptoms before the injury; iTMD, the patients who presented with idiopathic/non-traumatic TMD symptoms without any history of head/neck trauma; LPM, lateral pterygoid muscle; MPM, medial pterygoid muscle. The results were obtained via ^a^ chi-square test (two-sided), and ^b^: Fisher’s exact test. A *p*-value < 0.05 was considered significant. *: *p*-value < 0.05, **: *p*-value < 0.01. Significant results are in bold text.

**Table 6 jcm-10-01404-t006:** Associated abnormal MRI findings with increasing TMD index.

	wTMD						iTMD					
	VAS		PI		Neck PI		VAS		PI		Neck PI	
	r	*p*-Value	r	*p*-value	r	*p*-Value	r	*p*-Value	r	*p*-Value	r	*p*-Value
TMD index												
PI	0.174	0.252	1.000		**0.866 *****	<0.001	**0.300 ***	0.045	1.000		**0.764 *****	<0.001
Neck PI	0.081	0.597	**0.866 *****	<0.001	1.000		**0.301 ***	0.044	**0.764 *****	<0.001	1.000	
Abnormality of masticatory muscle												
Temp. VC	−0.124	0.416	−0.180	0.236	−0.006	0.970	−0.029	0.849	−0.059	0.703	−0.035	0.820
Temp. SC	−0.124	0.416	−0.180	0.236	−0.006	0.970	−0.029	0.849	−0.059	0.703	−0.035	0.820
LPM VC	−0.254	0.092	−0.089	0.563	0.094	0.540	−0.135	0.377	−0.210	0.167	−0.092	0.547
LPM SC	**0.346 ***	0.020	0.069	0.650	0.229	0.130	−0.185	0.224	0.061	0.689	−0.171	0.262
MPM VC	-	-	-	-	-	-	−0.217	0.153	−0.293	0.051	−0.064	0.674
MPM SC	-	-	-	-	-	-	−0.217	0.153	−0.293	0.051	−0.064	0.674
Abnormality of TMJ												
ADDWR	−0.043	0.781	0.208	0.171	0.130	0.395	0.064	0.674	−0.202	0.184	−0.178	0.241
ADDWoR	0.040	0.793	−0.158	0.299	−0.084	0.582	**0.325 ***	0.029	0.124	0.418	0.169	0.268
Disc deformity	−0.175	0.251	0.000	1.000	−0.029	0.852	0.200	0.187	−0.082	0.591	−0.007	0.962
Condylar degeneration	−0.053	0.730	−0.089	0.563	−0.144	0.345	**0.297 ***	0.047	−0.070	0.646	0.098	0.521
Effusion	0.021	0.891	−0.274	0.068	−0.134	0.379	0.180	0.236	0.181	0.234	0.067	0.664

TMD; temporomandibular disorder, wTMD; the patients had experienced whiplash injury and had no TMD symptoms before the injury, iTMD; the patients who had presented with idiopathic/non-traumatic TMD symptoms without any history of head/neck trauma, r: correlation coefficient, PI: palpation index, Temp.: temporal muscle, LPM: lateral pterygoid muscle, MPM; medial pterygoid muscle, VC: volume change, SC: signal change, ADDWR: anterior disc displacement with reduction, ADDWoR; anterior disc displacement without reduction. Volume and signal changes of masseter muscle were not observed at all, so the correlation coefficient related to this muscle could not be determined. The results were obtained via Spearman’s correlation analysis. A *p*-value < 0.05 was considered significant. **: *p*-value < 0.01, ***: *p*-value < 0.001. Significant results are in bold text.

## Data Availability

The datasets generated and/or analyzed during this study are available from the corresponding author upon reasonable request. Because patient consent is required for data disclosure, we may disclose data conditionally through internal discussion and the Institutional Review Board (IRB) of the Kyung Hee University Dental Hospital.

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
