# Peer review of "MRI-Based Assessment of Masticatory Muscle Changes in TMD Patients after Whiplash Injury"

_jcm, 2021, doi:10.3390/jcm10071404_

Round 1
Reviewer 1 Report
This MRI-based retrospective case-control study aimed to investigate the structural changes of masticatory muscles and TMJ in patients who suffered from whiplash-associated temporo-mandibular disorders in correlation to self-reported patient symptomatology, as compared to a control group of patients suffering from idiopathic non-traumatic temporo-mandibular disorders.
Overall, the study is interesting and of sound methodology, although quite similar to previous papers published by the authors (Lee et al. Front Neurol. 2017, 2018; Lee et al. J Oral Rehab. 2019). The results are limited but raise interesting questions and future perspectives. Nevertheless, significant changes must be made to the manuscript before it can be considered for publication in the Journal of Clinical Medicine.
Major issues:
- Methodological issues:
1 – The study design is not clear. It is presented as a retrospective case-control study of randomly selected patients, who all benefited from the same clinical examination protocol and assessments, without any missing data in either study groups. If the patients all undergo the same clinical examination protocol as part of their routine workup, thus allowing precise data gathering even retrospectively, this should be specified in the manuscript. In such case, how did you account for interobserver variability?
2 – How did you measure muscle volume changes/fibrosis as stated P5?
3 – In the second proposed hypothesis, the latter part, ie.: “… and differences between the wTMD and iTMD groups can be observed” seems quite self-evident for a case-control study and could most probably be removed.
- Scientific issues:
4 – Although statistically-significant differences between wTMD and iTMD patients are clearly evidenced in the present study, none of the results presented in the paper support the authors’ conclusion that “…patients with TMD who have whiplash macrotrauma can have different neurophysiological responses and pain control mechanisms than those with idiopathic causes.” (P15), especially in the absence of CPM experiments or central sensitization questionnaires. The paper’s conclusion is thus unsubstantiated and needs to be rewritten, limited to the observed changes in muscle volume and/or signal and pain symptomatology.
5 – There are several critical errors that need to be addressed:
P3 – The MRI characteristics of masticatory muscle changes as defined in the study by Zanoteli et al. 2002 pertain to myotonic dystrophy, which is a genetic muscular dystrophy, thus pathophysiologically different that TMD-related muscle disorders. It is thus wrong to extrapolate without further evidence of similar presentations between the two entities.
P7 – “The distributions of bruxism and tinnitus in the wTMD group were different from those in the iTMD group”. There were actually no differences in the groups for those two factors. Please correct.
P7 – The factors presented Table 2 are not (all) contributing factors. Please modify the title.
P14 – “Abnormal positioning of the articular disc may cause closed lock jaw position, which evokes TMJ clicking and pain, as well as limits jaw function [47].” In the case of closed lock jaw, the condyle cannot return to its normal position under the disk, thus not eliciting any clicking sounds upon mandibular movements. Please correct.
P15 – “Headache and sleep problems possibly co-occur from the result of the dysregulation of shared brain regions, such as the trigeminal nucleus caudalis and thalamus [62].” This is an erroneous oversimplification that needs to be corrected.
P15 – Stating that the study has a “randomized controlled design” is misleading. Please modify.
P15 – Although it is plausible that “…the pathological changes in one masticatory muscle will ultimately affect other muscle pathologies”, it is erroneously simplistic to justify such statement as follows: “Because the masticatory muscles function cooperatively and elaborately during jaw movement, [67]”. Indeed, there is increasing evidence to suggest that the functional coupling between masticatory muscles or between masticatory muscles and cervical muscles is mediated by central neural processes within the brainstem trigeminocervical complex and not a simple biomechanical coupling (see Devoize et al. Brain Res. 2010 or Giannakopoulos et al. J Oral Rehab. 2018 for details).
6 – The concept of “CNS-related factors” (P15) is quite unclear. From the statement “Finally, we observed that headache, sleep problems, and psychological distress were significantly more prevalent in the wTMD group than in the iTMD group. We must focus on the fact that CNS-related factors were more prevalent in the wTMD group.” (P15), these “CNS-related factors” seem akin to comorbidities. Nevertheless, in the same paragraph, such factors seem to imply changes in central pain processing. Please clarify this important issue.
7 – The discussion is quite unstructured, insufficiently focused and thus difficult to follow. Significant modifications need to be made:
- Summarizing the main results as answers to the hypotheses raised P3 will help bring clarity to the paper.
- Apart from discussing the muscles changes associated with whiplash, the discussion should focus on potential differences between the two groups in terms of central sensitization and nociplastic changes, that could explain the study results (see below).
- Considering that the wTMD group presented more structural changes (including more disc deformity and irreducible disk displacements [Table 3] and muscles changes [Table 4]) that correlated with the pain intensity, compared to the iTMD group (although there were fewer muscle changes in this group), the decorrelation between pain levels and structural aspects needs to be discussed. Discussing the role of putative nociplastic changes in the iTMD group would be interesting.
- The last paragraph of the discussion (P14) does not bring much to the discussion and can easily be removed.
8 – “Thus, patients with TMD suffering from whiplash injury have wider and stronger pain than those with only microtrauma.” (P2) – This sentence is unsubstantiated. Please add appropriate reference.
- Didactic issues:
9 – Several parts of the manuscript are unclear and/or difficult to follow, and thus need to be rewritten:
P1 – “It is important to note that patients commonly suffer from chronic whiplash injury and clinically have pain in a wider range of the body [8].” (P1)
P2 – The third paragraph (starting with “The neuropathology of TMD development…”) is difficult to follow. Please rewrite.
P1, P2, P13, P15 – The terms “neuropathophysiology” or “neuropathology” are improper in the present context. These should be replaced throughout the manuscript with the more appropriate “pathophysiology”.
10 – Figure 6 would gain in didactism if the 2 MRI images were paired with images from healthy control subjects to help in clearly visualizing the extent of volume/signal change in the LPM and disk displacement.
11 – The Institutional Review Board approval number needs to be provided in the manuscript. Please add.
Minor issues:
- Minor omissions:
12 – Table 4 is not called in the text. Please add.
13 – Figure 7 needs a unit legend to be understandable.
- Minor errors:
14 – In the beginning of the discussion (P13), whiplash is defined as a microtrauma. Please correct.
15 – P4 - There is an extra space in the third paragraph and an extra semicolon in the fourth.
- Formatting issues:
16 – There is a minor formatting error in Table 2.
17 – There is a formatting error at the end of reference 67.
Author Response
We wish you all healthy and enjoyable despite the ongoing COVID-19 pandemic.
We are very appreciated your thoughtful and reasonable suggestions and comments. We tried our best to correct or reflect your comments and suggestions, and sincerely want our research to be published in the Journal of Clinical Medicine.
Reviewer 1
Comments and Suggestions for Authors
This MRI-based retrospective case-control study aimed to investigate the structural changes of masticatory muscles and TMJ in patients who suffered from whiplash-associated temporo-mandibular disorders in correlation to self-reported patient symptomatology, as compared to a control group of patients suffering from idiopathic non-traumatic temporo-mandibular disorders.
Overall, the study is interesting and of sound methodology, although quite similar to previous papers published by the authors (Lee et al. Front Neurol. 2017, 2018; Lee et al. J Oral Rehab. 2019). The results are limited but raise interesting questions and future perspectives. Nevertheless, significant changes must be made to the manuscript before it can be considered for publication in the Journal of Clinical Medicine.
Response: We would like to sincerely thank you and the reviewers for comments and decisions. And we definitely want to publish our research in the Journal of Clinical Medicine, and we have worked hard to faithfully reflect the comments and suggestions of Reviewers throughout the manuscript. We have revised the text to fully reflect the opinions of you and the Reviewers, and the revised parts are marked in red. Thank you.
Major issues:
Methodological issues:
1 – The study design is not clear. It is presented as a retrospective case-control study of randomly selected patients, who all benefited from the same clinical examination protocol and assessments, without any missing data in either study groups. If the patients all undergo the same clinical examination protocol as part of their routine workup, thus allowing precise data gathering even retrospectively, this should be specified in the manuscript. In such case, how did you account for interobserver variability?
Response: Thank you very much for your thoughtful comments. Not all TMD patients who visited Kyung Hee University Dental Hospital within the set study period had all data, and the cases with any level of data loss were excluded from the study. To reflect your opinion and express it more clearly, we have revised the relevant part of the manuscript. We used Cohen's kappa coefficient to measure inter-rater reliability for the presence of MRI abnormalities. We have added this to the main body of the paper.
Not all TMD patients who visited our hospital within the
2 – How did you measure muscle volume changes/fibrosis as stated P5?
Response: Thank you very much for your question. We evaluated muscle atrophy, contracture, and morphological alterations in each masticatory muscle. The masticatory muscle was considered atrophic when fatty replacement tissue with a high-intensity signal was present across wide areas of the muscle on T1WIs and T2WIs. In the same images, contracture of the muscle presented as fibrosis and appeared as an area of low-density signal. This is explained in detail in our previous paper [1], and we have chosen the same evaluation method.
In order to clarify the meaning of the sentence and to reflect your opinion, we have revised the corresponding sentence as follows.
→ In the same images, it was considered to have a VC when the masticatory muscle had decreased volume and fibrosis which had the muscle contracture with low-density signal.
3 – In the second proposed hypothesis, the latter part, ie.: “… and differences between the wTMD and iTMD groups can be observed” seems quite self-evident for a case-control study and could most probably be removed.
Response: Thank you for your sincere comments. We have deleted the part you pointed out and corrected the area as follows.
(2) structural changes in the TMJ and masticatory muscles will be associated with a higher severity of TMD symptoms, and differences between the wTMD and iTMD groups can be observed.
→ (2) structural changes in the TMJ and masticatory muscles will be associated with a higher severity of TMD symptoms.
Scientific issues:
4 – Although statistically-significant differences between wTMD and iTMD patients are clearly evidenced in the present study, none of the results presented in the paper support the authors’ conclusion that “…patients with TMD who have whiplash macrotrauma can have different neurophysiological responses and pain control mechanisms than those with idiopathic causes.” (P15), especially in the absence of CPM experiments or central sensitization questionnaires. The paper’s conclusion is thus unsubstantiated and needs to be rewritten, limited to the observed changes in muscle volume and/or signal and pain symptomatology.
Response: We found your opinion very reasonable, so we have decided to accept your opinion. However, in our results, the wTMD group showed higher VAS, PI, and neck PI scores than the iTMD group. In addition, we hope that you will reconsider that wTMD group had a higher frequency of headache, sleep problem, and psychological stress.
The sentence you pointed out was modified as follows.
We suggest that patients with TMD who have whiplash macrotrauma can have different neurophysiological responses and pain control mechanisms than those with idiopathic causes.
→ We suggest that patients with TMD who have whiplash macrotrauma can have differentiated pain symptomatology from those with idiopathic causes, having more pain in a wider area of the body and experiencing more volume and signal changes in the masticatory muscle.
5 – There are several critical errors that need to be addressed:
P3 – The MRI characteristics of masticatory muscle changes as defined in the study by Zanoteli et al. 2002 pertain to myotonic dystrophy, which is a genetic muscular dystrophy, thus pathophysiologically different that TMD-related muscle disorders. It is thus wrong to extrapolate without further evidence of similar presentations between the two entities.
Response: Thank you for your valuable and sincere comment, we have decided to accept your opinion. We changed it to an appropriate reference.
P7 – “The distributions of bruxism and tinnitus in the wTMD group were different from those in the iTMD group”. There were actually no differences in the groups for those two factors. Please correct.
Response: Thank you very much for your comment. We have modified the sentence you pointed out as follows.
The distributions of bruxism and tinnitus in the wTMD group were different from those in the iTMD group.
→ The distributions of bruxism and tinnitus in the wTMD group were not significantly different from those in the iTMD group.
P7 – The factors presented Table 2 are not (all) contributing factors. Please modify the title.
Response: Thank you for valuable comment. We have modified the part you pointed out as follows.
Table 2. Distribution of contributing factors
→ Table 2. Distribution of contributing factors and comorbidities of TMD
P14 – “Abnormal positioning of the articular disc may cause closed lock jaw position, which evokes TMJ clicking and pain, as well as limits jaw function [47].” In the case of closed lock jaw, the condyle cannot return to its normal position under the disk, thus not eliciting any clicking sounds upon mandibular movements. Please correct.
Response: Thank you very much for sincere comments. We have modified the part you pointed out as follows.
Abnormal positioning of the articular disc may cause closed lock jaw position, which evokes TMJ clicking and pain, as well as limits jaw function.
→ Abnormal positioning of the articular disc can evoke TMJ clicking and pain, can lead to triggering a closed locking jaw position, or may as well as limit jaw function.
P15 – “Headache and sleep problems possibly co-occur from the result of the dysregulation of shared brain regions, such as the trigeminal nucleus caudalis and thalamus [62].” This is an erroneous oversimplification that needs to be corrected.
Response: Thank you for your thoughtful and reasonable comments. We modified the sentence as follows:
Headache and sleep problems possibly co-occur from the result of the dysregulation of shared brain regions, such as the trigeminal nucleus caudalis and thalamus.
→ Headache and sleep problems possibly co-occur from the shared underlying mechanisms result of reducing nociceptive activity in trigeminal nucleus caudalis or increasing dysfunctional hypothalamic activity [2,3].
P15 – Stating that the study has a “randomized controlled design” is misleading. Please modify.
Response: Thank you for your thoughtful comments. And we have changed the part to “retrospective case-control study”. Thank you.
P15 – Although it is plausible that “…the pathological changes in one masticatory muscle will ultimately affect other muscle pathologies”, it is erroneously simplistic to justify such statement as follows: “Because the masticatory muscles function cooperatively and elaborately during jaw movement, [67]”. Indeed, there is increasing evidence to suggest that the functional coupling between masticatory muscles or between masticatory muscles and cervical muscles is mediated by central neural processes within the brainstem trigeminocervical complex and not a simple biomechanical coupling (see Devoize et al. Brain Res. 2010 or Giannakopoulos et al. J Oral Rehab. 2018 for details).
Response: Thank you very much for your comments and suggestions. We sincerely thank you for your wise suggestion, and we have tried to fully reflect it. Thank you.
→ In addition, the functional association between the masticatory muscles or between masticatory muscles and cervical muscles is mediated by complex central neural processes within the brainstem trigeminocervical complex and not a simple biomechanical coupling [69,70].
6 – The concept of “CNS-related factors” (P15) is quite unclear. From the statement “Finally, we observed that headache, sleep problems, and psychological distress were significantly more prevalent in the wTMD group than in the iTMD group. We must focus on the fact that CNS-related factors were more prevalent in the wTMD group.” (P15), these “CNS-related factors” seem akin to comorbidities. Nevertheless, in the same paragraph, such factors seem to imply changes in central pain processing. Please clarify this important issue.
Response: Thank you for your comments and suggestions. We hypothesized that, unlike habits such as clenching and bruxism and that occur at the ends of the body, headache [4], sleep problem [5], and psychological distress [6] may be more correlated with events occurring at the CNS-level. Therefore, I have corrected the term you pointed out from CNS-related factors to comorbidities associated with CNS involvement.
→ Finally, we observed that headache, sleep problems, and psychological distress were significantly more prevalent in the wTMD group than in the iTMD group. Headaches, sleep problems, and psychological distress may be related to CNS pathology rather than peripheral pathologies [4,6,7]. We must focus on the fact that comorbidities associated with CNS involvement were more prevalent in the wTMD group. Headache and sleep problems possibly co-occur from the shared underlying mechanisms result of reducing nociceptive activity in trigeminal nucleus caudalis or dysfunctional hypothalamic activity [2,3].
7 – The discussion is quite unstructured, insufficiently focused and thus difficult to follow. Significant modifications need to be made:
Summarizing the main results as answers to the hypotheses raised P3 will help bring clarity to the paper.
Response: In your opinion, the authors have gone through several virtual meetings and revised the Discussion section to be more structured. Thanks for the attentive and reasonable advice. We made a full revision of the Discussion section, and we hope you read the red section of the manuscript carefully. We did our best. Thank you.
→ As in our first hypothesis, patients with wTMD had higher pain intensity across wider jaw and neck areas compared to those with iTMD. This was consistent with previous results that patients with whiplash injury report higher pain scores and larger areas of local and referred pain than healthy controls [8].
Apart from discussing the muscles changes associated with whiplash, the discussion should focus on potential differences between the two groups in terms of central sensitization and nociplastic changes, that could explain the study results (see below).
Response: Thank you very much for your suggestions. The manuscript has been revised to reflect your advice.
→ As in our first hypothesis, patients with wTMD had higher pain intensity across wider jaw and neck areas compared to those with iTMD. This was consistent with previous results that patients with whiplash injury report higher pain scores and larger areas of local and referred pain than healthy controls [8]. The increased clinical pain observed in the wTMD group can be related to nociplastic pain, which implies changes in function of nociceptive pathways [9]. This pain pattern can be caused by insufficient or impaired diffuse noxious inhibitory controls (DNICs), a measure of central nervous system pain inhibition [10].
→ Interestingly, central sensitization is not only specific to whiplash injury, even though this phenomenon is seen in a variety of chronic pain syndromes [11,12].
Considering that the wTMD group presented more structural changes (including more disc deformity and irreducible disk displacements [Table 3] and muscles changes [Table 4]) that correlated with the pain intensity, compared to the iTMD group (although there were fewer muscle changes in this group), the decorrelation between pain levels and structural aspects needs to be discussed. Discussing the role of putative nociplastic changes in the iTMD group would be interesting.
Response: Thank you for your sincere and smart suggestions. The manuscript has been fully revised to reflect your advice. Actually, we have written a new paragraph. I ask for your confirmation. Thank you.
The last paragraph of the discussion (P14) does not bring much to the discussion and can easily be removed.
Response: Thank you for your suggestions. We have modified the section you pointed out.
8 – “Thus, patients with TMD suffering from whiplash injury have wider and stronger pain than those with only microtrauma.” (P2) – This sentence is unsubstantiated. Please add appropriate reference.
Response: Thank you very much for your comments. We have added the appropriate reference.
→ Thus, patients with TMD suffering from whiplash injury have wider and stronger pain than those with only microtrauma [13].
Didactic issues:
9 – Several parts of the manuscript are unclear and/or difficult to follow, and thus need to be rewritten:
Response: Thank you for your sincere comments.
P1 – “It is important to note that patients commonly suffer from chronic whiplash injury and clinically have pain in a wider range of the body [8].” (P1)
Response: We have revised the sentence as follows to convey the clear meaning of the sentence.
→ It is important to note that patients suffering from chronic whiplash injury commonly have clinical pain in a wider range of their bodies.
P2 – The third paragraph (starting with “The neuropathology of TMD development…”) is difficult to follow. Please rewrite.
Response: Thank you for your comments. We have rewritten the paragraphs pointed out.
P1, P2, P13, P15 – The terms “neuropathophysiology” or “neuropathology” are improper in the present context. These should be replaced throughout the manuscript with the more appropriate “pathophysiology”.
Response: Thank you for your comments. We have change the terms “neuropathophysiology” or “neuropathology” to “pathophysiology”. Thank you.
10 – Figure 6 would gain in didactism if the 2 MRI images were paired with images from healthy control subjects to help in clearly visualizing the extent of volume/signal change in the LPM and disk displacement.
Response: Thank you for your comments. We have added the 2 MRI images from healthy control subjects. Thank you.
11 – The Institutional Review Board approval number needs to be provided in the manuscript. Please add.
Response: Thank you for your kind comments. We have added the Institutional Review Board approval number. Thank you.
Minor issues:
Minor omissions:
12 – Table 4 is not called in the text. Please add.
Response: Thank you for your kind comments. We have added Table 4 in the text.
13 – Figure 7 needs a unit legend to be understandable.
Response: Thank you for your comments. We have added a unit legend.
Minor errors:
14 – In the beginning of the discussion (P13), whiplash is defined as a microtrauma. Please correct.
Response: Thanks for your meticulous comment, and this part has been modified with microtrauma.
15 – P4 - There is an extra space in the third paragraph and an extra semicolon in the fourth.
Response: Thanks for your meticulous comment.
Formatting issues:
16 – There is a minor formatting error in Table 2.
Response: Thanks for your comment. We have modified the formatting error in Table 2.
17 – There is a formatting error at the end of reference 67.
Response: Thank you very much for your comment. We have modified it.
References
- Lee, Y.H.; Lee, K.M.; Auh, Q.S.; Hong, J.P. Magnetic Resonance Imaging-Based Prediction of the Relationship between Whiplash Injury and Temporomandibular Disorders. Front Neurol 2017, 8, 725, doi:10.3389/fneur.2017.00725.
- Sullivan, D.P.; Martin, P.R.; Boschen, M.J. Psychological Sleep Interventions for Migraine and Tension-Type Headache: A Systematic Review and Meta-Analysis. Sci Rep 2019, 9, 6411-6411, doi:10.1038/s41598-019-42785-8.
- Brennan, K.C.; Charles, A. Sleep and headache. Semin Neurol 2009, 29, 406-418, doi:10.1055/s-0029-1237113.
- La Mantia, L.; Erbetta, A. Headache and inflammatory disorders of the central nervous system. Neurol Sci 2004, 25 Suppl 3, S148-153, doi:10.1007/s10072-004-0275-7.
- Provini, F.; Lombardi, C.; Lugaresi, E. Insomnia in neurological diseases. Semin Neurol 2005, 25, 81-89, doi:10.1055/s-2005-867074.
- de Morree, H.M.; Szabó, B.M.; Rutten, G.J.; Kop, W.J. Central nervous system involvement in the autonomic responses to psychological distress. Neth Heart J 2013, 21, 64-69, doi:10.1007/s12471-012-0351-1.
- Dyken, M.E.; Afifi, A.K.; Lin-Dyken, D.C. Sleep-related problems in neurologic diseases. Chest 2012, 141, 528-544, doi:10.1378/chest.11-0773.
- Johansen, M.K.; Graven-Nielsen, T.; Olesen, A.S.; Arendt-Nielsen, L. Generalised muscular hyperalgesia in chronic whiplash syndrome. PAIN 1999, 83.
- Freynhagen, R.; Parada, H.A.; Calderon-Ospina, C.A.; Chen, J.; Rakhmawati Emril, D.; Fernández-Villacorta, F.J.; Franco, H.; Ho, K.Y.; Lara-Solares, A.; Li, C.C., et al. Current understanding of the mixed pain concept: a brief narrative review. Curr Med Res Opin 2019, 35, 1011-1018, doi:10.1080/03007995.2018.1552042.
- Pereira-Silva, R.; Costa-Pereira, J.T.; Alonso, R.; Serrão, P.; Martins, I.; Neto, F.L. Attenuation of the Diffuse Noxious Inhibitory Controls in Chronic Joint Inflammatory Pain Is Accompanied by Anxiodepressive-Like Behaviors and Impairment of the Descending Noradrenergic Modulation. Int J Mol Sci 2020, 21, doi:10.3390/ijms21082973.
- Sterling, M.; Jull, G.; Vicenzino, B.; Kenardy, J. Sensory hypersensitivity occurs soon after whiplash injury and is associated with poor recovery. Pain 2003, 104, 509-517, doi:10.1016/s0304-3959(03)00078-2.
- Sterling, M. Whiplash-associated disorder: musculoskeletal pain and related clinical findings. J Man Manip Ther 2011, 19, 194-200, doi:10.1179/106698111X13129729551949.
- Lee, Y.-H.; Lee, K.M.; Auh, Q.S.; Hong, J.-P. Magnetic Resonance Imaging-Based Prediction of the Relationship between Whiplash Injury and Temporomandibular Disorders. Frontiers in neurology 2018, 8, 725-725, doi:10.3389/fneur.2017.00725.

Reviewer 2 Report
The specificity of MRI needs to show a homogeneity in the sample. It should be explained well how long the patients suffered from TMD, because there is a difference among those who have been suffering from TMD for 10 years and those who have been suffering from it for only 1year.
The timing of MRI examination of the trauma should be more clearly explained in order to understand if the only difference between the two groups was caused by whiplash.
The closed Lock TM is an evolution of a tm incoordination, and from what we can see the two RM figures 4 ab and 6 a show a reworking of the upper condylar pole which represents an old alteration.
Only in this way can the work be taken into consideration.
Author Response
We wish you all healthy and enjoyable despite the ongoing COVID-19 pandemic.
We are very appreciated your thoughtful and reasonable suggestions and comments. We tried our best to correct or reflect your comments and suggestions and sincerely want our research to be published in the Journal of Clinical Medicine. We have marked the revised parts in red (manuscript).
Reviewer 2
Comments and Suggestions for Authors
The specificity of MRI needs to show a homogeneity in the sample. It should be explained well how long the patients suffered from TD, because there is a difference among those who have been suffering from TMD for 10 years and those who have been suffering from it for only 1year.
Response: We deeply appreciate your reasonable suggestion and thank you very much for your comment. We presented the symptom duration in Table 1. In our study, wTMD group was consisted with the patients had first-onset TMD after whiplash injury [1,2]. As you mentioned, there may have been changes to the TMJ and surrounding structures before the whiplash trauma, but in reality we haven't been able to investigate all of these. Clinically, we can only consider the timing of the onset of symptoms that the patient reports to the doctor, rather than the timing of the onset of histological changes, such as disc displacement or muscle inflammation observed in images. The reason is that patients do not come to the hospital when they do not have clinical symptoms, and they must have recognizable jaw joint symptoms to come to the hospital. In other words, it is not possible to know exactly when the patient's TMD sign began. We assumed that the time when a patient visits the hospital to diagnose his or her TMD condition and begins to recognize the symptoms they want to treat is usually the 'onset' point. Unfortunately, this study is not a cohort design, which may lead to limitations of the method. We promise that we will take into account the points you have pointed out in subsequent research.
The timing of MRI examination of the trauma should be more clearly explained in order to understand if the only difference between the two groups was caused by whiplash.
Response: Thanks for your wise advice. Thanks for your wise advice. We are taking MRI scans at the beginning of the patient's visit to our university hospital. Therefore, the following sentence has been added to more clearly express the condition.
→ MRIs for patients with TMD were taken during the TMD diagnosis phase, between the first and second hospital visits.
The closed Lock TM is an evolution of a tm incoordination, and from what we can see the two RM figures 4 ab and 6 a show a reworking of the upper condylar pole which represents an old alteration.
Response: We sincerely appreciate your comments. In fact, patients with whiplash injury related TMD were admitted to the hospital within a few days to 4 months of the injury. We don't know if the TMD we had before that has worsened, or if the TMJ and surrounding tissues have changed within a short period of time. It only presents the representative example observed as an image. Just know we did our best. We sincerely thank you for the opportunity.
Only in this way can the work be taken into consideration.
Response: Thank you very much for your comment.
References
- Slade, G.D.; Bair, E.; Greenspan, J.D.; Dubner, R.; Fillingim, R.B.; Diatchenko, L.; Maixner, W.; Knott, C.; Ohrbach, R. Signs and symptoms of first-onset TMD and sociodemographic predictors of its development: the OPPERA prospective cohort study. J Pain 2013, 14, T20-32.e21-23, doi:10.1016/j.jpain.2013.07.014.
- Lee, Y.-H.; Lee, K.M.; Auh, Q.S.; Hong, J.-P. Magnetic Resonance Imaging-Based Prediction of the Relationship between Whiplash Injury and Temporomandibular Disorders. Frontiers in neurology 2018, 8, 725-725, doi:10.3389/fneur.2017.00725.
Round 2
Reviewer 1 Report
The requests changes have been made. Minor english corrections to the added sections of the revised manuscript will improve the clarity of the manuscript, but these can be done at the proofreading stage.
The manuscript is otherwise suitable for publication.
Reviewer 2 Report
Dear authors, I am satisfied with your answers. It is an interesting scientific work. Best regards